# Metabolic Syndrome, Thyroid Dysfunction, and Cardiovascular Risk: The Triptych of Evil

**DOI:** 10.3390/ijms251910628

**Published:** 2024-10-02

**Authors:** Alessandro Pingitore, Melania Gaggini, Francesca Mastorci, Laura Sabatino, Linda Cordiviola, Cristina Vassalle

**Affiliations:** 1Clinical Physiology Institute, CNR, 56124 Pisa, Italy; melania.gaggini@cnr.it (M.G.); francesca.mastorci@cnr.it (F.M.); laura.sabatino@cnr.it (L.S.); 2Department of Pharmacy, Pisa University, 56126 Pisa, Italy; l.cordiviola1@studenti.unipi.it; 3Fondazione G. Monasterio, Regione Toscana, 56124 Pisa, Italy; cricca@ftgm.it

**Keywords:** metabolic syndrome, thyroid hormones, thyroid dysfunction, cardiovascular risk, endothelial dysfunction, atherosclerosis

## Abstract

The triad formed by thyroid dysfunction, metabolic syndrome (MetS), and cardiovascular (CV) risk forms a network with many connections that aggravates health outcomes. Thyroid hormones (THs) play an important role in glucose and lipid metabolism and hemodynamic regulation at the molecular level. It is noteworthy that a bidirectional association between THs and MetS and their components likely exists as MetS leads to thyroid dysfunction, whereas thyroid alterations may cause a higher incidence of MetS. Thyroid dysfunction increases insulin resistance, the circulating levels of lipids, in particular LDL-C, VLDL-C, and triglycerides, and induces endothelial dysfunction. Furthermore, THs are important regulators of both white and brown adipose tissue. Moreover, the pathophysiological relationship between MetS and TH dysfunction is made even tighter considering that these conditions are usually associated with inflammatory activation and increased oxidative stress. Therefore, the role of THs takes place starting from the molecular level, then manifesting itself at the clinical level, through an increased risk of CV events in the general population as well as in patients with heart failure or acute myocardial infarction. Thus, MetS is frequently associated with thyroid dysfunction, which supports the need to assess thyroid function in this group, and when clinically indicated, to correct it to maintain euthyroidism. However, there are still several critical points to be further investigated both at the molecular and clinical level, in particular considering the need to treat subclinical dysthyroidism in MetS patients.

## 1. Introduction

The relationship between metabolic syndrome (MetS), thyroid dysfunction (both hypothyroidism and hyperthyroidism), and cardiovascular (CV) risk is complex and multifaceted and involves several molecular mechanisms. Atherosclerosis, the leading cause of death from CV disease, is a chronic inflammatory condition, characterized by the accumulation of lipids and leukocytes in blood vessels, leading to plaque formation. Over time, a plaque hardens, causing narrowing of the arteries that limits blood flow, and when fatty plaques rupture, a thrombus (blood clot) can form, further blocking oxygen-rich blood flow to the body’s organs. MetS is a cluster of metabolic disorders characterized by the co-occurrence of various cardiometabolic risk factors, such as hypertension, increased fasting glucose, reduced high-density lipoprotein cholesterol levels, high triglycerides concentrations, and central obesity (accumulation of fat in the abdomen), that in turn can increase the risk of atherogenic damage [1]. Actually, according to the diagnostic criteria of MetS proposed by different health organizations, a general and unanimous consensus on diagnostic definition of MetS has not yet been reached [1,2,3,4]. According to the Adult Treatment Panel III (ATPIII) of the US National Cholesterol Education Program, the presence of any of the three abovementioned components are necessary for MetS diagnosis [4]. In general, MetS prevalence is increasing all over the word, including both in youths and adults [5]. Accordingly, the NHANES data showed a MetS prevalence increase from 36 to 47% in the period between 1999 and 2018 [6]. In the context of the MetS effects on the CV system, abnormalities of subclinical functional cardiac parameters, including left ventricular (LV) diastolic relaxation, even in the absence of hypertension, and systolic function assessed with echocardiographic strain have been detected [7,8]. In an Italian multicenter study, enrolling 6.422 asymptomatic MetS patients, diastolic left ventricular dysfunction was present in 45% and systolic LV dysfunction in 12.5% [8]. Moreover, in patients with coronary artery disease (CAD), MetS was independently associated with impairment of LV systolic deformation strain in addition to CAD as assessed with cardiac magnetic resonance future tracking [9]. Moreover, endothelial dysfunction is an early sign in MetS, contributing to CAD development and progression [10]. Contextually, coronary flow reserve is reduced in MetS patients [10]. A consequence of these dangerous effects of MetS on coronary circulation is the negative prognostic impact with the evidence of a high prevalence of cardiac events. This was recently confirmed in a meta-analysis including 10 studies and 33,197 patients with myocardial infarction. Patients with MetS had a higher prevalence of major CV events with a risk ratio of 1.35 [11]. Interestingly, in another study, MetS patients with diabetes had the highest long-term mortality and those obese without diabetes the lowest long-term mortality, suggesting that MetS does not represent a homogeneous group of patients [12,13]. Thyroid hormones (THs) play a key role in regulating the rate of carbohydrate and lipid metabolism, inducing also abnormalities of blood pressure values in the presence of thyroid metabolism disturbances. However, the relationship between MetS and TH disorders is still not clearly defined, and the current results are contrasting. Accordingly, a recent meta-analysis did not raise any conclusion to high heterogeneity in reporting results [14]. Another study showed that each unit increase in TSH was associated with a 3% increase in the odds of prevalent MetS. When considering subclinical hypothyroidism (SCH) with a TSH > 10 mIU/L, an increased odds of prevalent but not incident MetS has been shown [15]. In addition, an age factor has been shown by Wu et al. with the evidence of a relationship of SHYPO and MetS in young men [16]. Triiodothyronine (T3) is considered the thyroid hormone in the active form and derives from 5′-monodeiodination of thyroxine (T4), produced directly by the thyroid gland. This process, occurring in the peripheral tissues, is mediated by type I (DIO1) and type II (DIO_2_) deiodinases. T3 acts on the heart through genomic and non-genomic mechanisms. The genomic ones are mediated by thyroid receptors (TRs) α1 and α2, and β1 and β2. They bind to TH response elements (TREs) in the promoter region of genes. In the heart, TRα1 is the more diffuse TR isoform and has a functional regulatory role, whereas TRα2 acts as a counter-regulator of TRα1, suppressing its transcriptional effects. The non-genomic actions are mediated through cytoplasmic and membrane-associated TRs, such as integrin αVβ3 or extranuclear TRα and β, and involve ions, glucose, and amino acid transport across the plasma membrane. The TH effects on the CV system are also indirect through the regulation of hormonal or neuroendocrine pathways [17]. Thus, through these mechanisms, THs exert effects on cardiac morphology and structure, coronary vasculature, cell metabolism, cell protection, growth, and differentiation. The important CV role of THs is evident since mild thyroid dysfunctions (subclinical hypothyroidism or hyperthyroidism) induce changes in cardiac function and morphology [18] and are associated with increased CV morbidity and mortality [19,20,21,22,23,24,25]. In a recent study, TH metabolic abnormalities increased CV risk in the general population [26]. Accordingly, the 10-year absolute CV risk increased over 5% for women with free-T4 greater than the 85th percentile (median 17.6 pmol/L) and men with free-T4 greater than the 75th percentile (median 16.7 pmol/L). Similarly, low TSH circulating levels were associated with a higher risk of all-cause and CV mortality. In the context of heart failure and acute myocardial infarction, an abnormal TH profile has been associated with a high prevalence of major cardiac event, representing an independent predictor of cardiac death in addition to the common clinical and cardiac functional parameters [27,28]. Beyond the considerable direct effects on the heart, thyroid dysfunction may adversely influence the cardiometabolic system at several levels, calling into question dyslipidemia, endothelial dysfunction, hypertension, reduced flow-mediated dilation, and a series of events that contribute to the onset and development of the atherosclerotic plaque, including platelets and monocyte adhesion, LDL-oxidation, expression of thrombogenic factors, and migration and proliferation of smooth muscle cells [29,30]. These effects are much more evident in the presence of severe thyroid disturbances and are reversible when euthyroidism is restored. In this context, the intermingled relationship between THs, cardiometabolic risk is highlighted also in the recent SARS-CoV-2 pandemic; in fact, evidence emerged that preexisting traditional cardiovascular risk factors may negatively impact the risk and severity of COVID-19, as SARS-CoV-2 infection seems to affect the function of thyroid gland by different mechanisms [31,32,33,34].

With regard to the dysthyroid treatment, in the case of hypothyroidism, L-T4 substitutive treatment is recommended when TSH levels are higher than 10 mIU/L, starting with a dose of 20–25 mcg/day [35]. In the case of hyperthyroidism, it can be treated with antithyroid drugs, radioactive iodine ablation, or surgery. In a recent study, surgery was associated with lower long-term risks of cardiac events and all-cause mortality, while radioactive iodine ablation was associated with a lower incidence of cardiac events in comparison to antithyroid drugs [36]. The pathophysiological relationship between MetS, CV, and TH dysfunction is made even tighter considering that all conditions are associated with inflammatory activation and increased oxidative stress. This review approaches TH abnormalities, CV disease, and MetS according to molecular perspective and clinical implications.

## 2. TH, Insulin Resistance, and Cardiometabolic Risk

There are several experimental and clinical studies showing a strict bidirectional relationship between insulin and THs [37]. In the study of Ferrannini et al., higher levels of free-T3 were predictors of insulin resistance both cross-sectionally and longitudinally in euthyroid subjects even after adjusting for common risk factors of insulin resistance [38]. In patients with SCH, fasting hyperinsulinemia may precede the appearance of insulin resistance [39]. However, in another study, insulin levels and Homeostasis Model Assessment of Insulin Resistance (HOMA-IR) were elevated in patients with SCH in comparison to euthyroids, with TSH levels correlating directly with insulin and HOMA-IR, wherein free-T3 levels correlated negatively and strongly with insulin and moderately with HOMA-IR [40]. Moreover, Chubb et al. showed that the interaction between thyroid function and insulin sensitivity is an important contributor to diabetic dyslipidemia [41]. In the experimental setting, among the complex context of the cellular signaling network activated by insulin and THs, a critical molecular node of both insulin and TH mechanisms is the activation of the PI3K-AKT pathway [42]. Interestingly, TH induces proinsulin gene expression by the PI3K-AKT pathway, wherein hypothyroidism reduces proinsulin mRNA content that is restored with T3 treatment [43]. On one hand, this signaling induces a cascade of actions that regulates most of the metabolic actions of insulin, including the following: (1) glycogen synthesis through the inactivation by phosphorylation of glycogen synthase kinase 3 (GSK3), (2) glucose uptake through the phosphorylation and inhibition of the Rab-GTPase-activating protein (AS160) [44], (3) gluconeogenesis through the control of the winged or forkhead (FOX) class of transcription factors [45]. Interestingly, AKT signal activates the Mechanistic Target of Rapamycin (mTOR) pathway through the phosphorylation and consequent inhibition of the tuberin (known as tuberous sclerosis complex, TSC2), that is an mTOR inhibitor [46]. The mTOR pathway is an important regulator of protein synthesis and has been implicated in the T3-mediated antiapoptotic protection of the pancreatic beta-cells [47]. On the other hand, T3 activates the PI3K signaling pathway through nongenomic mechanisms, mainly by interacting with the integrin receptor αVβ3 on the plasma membrane [48]. TH effects mediated by the activation of PI3K-AKT signal have a relevant role in the CV protection. Interestingly, Carrillo-Sepulveda et al. showed that T3 induced nitric oxide (NO) production through AKT signal activation [49], and Pantos et al. showed the cardioprotective effect of TH in the context of ischemic cardiac disease [50]. Some other data, obtained in an insulin-resistant model, showed that T3 treatment in cardiac tissue may improve glucose metabolism, thus reducing the adverse effects of diabetes and MetS [51]. It is noteworthy to observe a TH dose-dependent effect on the AKT activation in the context of cardioprotection [52,53]. Interestingly, it was observed that a mild activation of AKT by TH administration to experimental animal induced cardioprotection, whereas elevated activation of AKT signaling by higher TH doses increased animal mortality. This study may be of relevant therapeutic importance because it shows that high doses of TH may be detrimental rather than beneficial.

## 3. TH, Dyslipidemia, and Cardiometabolic Risk

The systemic inflammatory process associated with MetS has numerous detrimental effects, promoting plaque formation and contributing to clinical events. Interactions between the innate immune system (white blood cells known as leukocytes) and lipid-derived products play a primary role in the pathophysiology of atherosclerosis. Individuals with MetS have an increased risk of developing atherosclerosis if compared to non-affected individuals, and the risk of CV disease increases when multiple components of MetS are present. THs exert their actions on all major metabolic pathways. With specific regard to lipid metabolism, THs affect synthesis, mobilization, and, even more, degradation of lipids. TRβ1 is the major TR in the liver and it is involved in lipid metabolism. THs and their derivatives that bind TRβ selectively or exhibit liver-selective uptake improve plasma lipids without affecting heart rate [54]. TRβ1 resulted as the most important mediator of the effects of T3 on cholesterol and lipoprotein metabolism in mice [55]. In several animal studies, treatment with TRβ1-selective TH analogs resulted in decreased plasma TG levels both in hypothyroid [56] and euthyroid mice [54]. In hypothyroidism, hyperlipidemia occurs and is characterized by elevated levels of Low-Density Lipoproteins (LDLs), very-low-density lipoproteins (VLDLs) and triglycerides. The progression of LDL particles into the sub-endothelium is involved in the formation of atherosclerotic plaques, contributing to atherosclerosis-related CV diseases. Modulatory biomarkers such as proprotein convertase subtilisin/kexin type 9 (PCSK9), angiopoietin-like protein (ANGPTL), and fibroblast growth factors (FGFs) play a significant role in modulating the risk of hypothyroidism-associated hyperlipidemia [57]. Additionally, under hypothyroid conditions, dysfunctional high-density lipoprotein (HDL) particles with reduced ability to promote reverse cholesterol transport (RCT) are observed [58]. In a multicenter retrospective study, patients with CAD were studied to determine the association between sensitivity to THs and TG, LDL, and TC levels, highlighting the importance of the pituitary–thyroid–heart axis in lipid metabolism susceptibility. The results showed that 68.34% of patients had dyslipidemia, and, in particular, central TH sensitivity was evaluated by the thyroid feedback quantile-based index (TFQI), parametric thyroid feedback quantile-based index (PTFQI), thyroid-stimulating hormone index (TSHI), and thyrotropin thyroxine resistance index (TT4RI). The ratio of free-T3/free-T4 was used to assess peripheral TH sensitivity. Multivariate logistic regression analysis confirmed that all TH sensitivity indices (TFQI, PTFQI, TSHI, TT4RI) were positively associated with dyslipidemia risk, and the free-T3/free-T4 level was negatively associated with dyslipidemia risk. Moreover, most associations were observed across sexes, glucose levels, and blood pressure groups when considered individually [59]. SCH is associated with adverse outcomes in terms of CV risk, quality of life, and metabolic condition. Recently, a study conducted by Chaudhary et al. aimed to find the prevalence of SCH in patients with myocardial infarction between patients with and without SCH. In this study, clinical examination, including thyroid status and lipid profile (LDL, HDL, triglycerides, and glycerol) evaluation, was performed. The results showed that patients with SCH and myocardial infarction (MI) had significant dyslipidemia when compared to euthyroid subjects, which could be a determining factor in the progression to MI [60]. Demirhan et al. investigated the effects of hypothyroidism on Non-Alcoholic Fatty Liver Disease and atherosclerosis using non-invasive indices of steatosis and fibrosis as AST-to-Platelet Ratio Index (APRI) Score, Fibrosis-4 (FIB-4), and atherogenic index of plasma (API). In 1370 patients with hypothyroidism (85.4% were female), high AIP values were associated with an increased CV risk. AIP increased as TSH increased in hypothyroid patients, whereas no significant gain was detected in APRI and FIB-4 evaluating NAFLD. The rise in AIP with TSH values suggests that high TSH values in hypothyroidism promote dyslipidemia, thus favoring the development or progression of atherosclerosis [61]. Thus, thyroid evaluation could be strongly recommended in patients with metabolic alterations; drug-inducing thyroid dysfunction should be carefully monitored, especially in patients where TH levels are altered [62].

## 4. THs and Adipose Tissue: Which Role in Obesity and Browning?

The adipose tissue, a target of THs, in virtue of its main role of energy storage, acts as a regulator of energy balance, sending signals to maintain metabolic control. Furthermore, the adipose tissue is important in the transport, synthesis, and mobilizations of lipids in conditions of increased demand, such as fasting, cold, and stress in general [63]. Commonly, the adipose tissue is divided into white (WAT) and brown (BAT) adipose tissue, structurally and functionally defined: white fat cells are important for chemical energy storage, whereas brown adipocytes have the role to protect mammals from hypothermia, obesity, and diabetes utilizing high levels of mitochondrial UCP1 to uncouple respiration and dissipate energy as heat. This is particularly evident in rodents and other small animals, which retain large amounts of BAT all their life, whereas larger animals reduce brown fat depots after infancy. Recently, developed Positron emission tomography (PET)–scanning technologies indicated the presence in adult humans of significant deposits of brown fat cells positive to uncoupling protein-1 (UCP-1), present in the inner membrane of mitochondria, principally in the upper trunk (i.e., in cervical, supraclavicular, paravertebral, pericardial areas). WAT and BAT are both targets of THs since the TR isoforms TRα1, TRα2, and TRβ1 are present, with a prevalence of TRα1 [64]. TH effects on adipocytes depend on the intracellular levels of T3, formed by deiodination of T4 by DIO1 or DIO_2_ [65]. No definitive data are currently available on the involvement of deiodinases at the adipose tissue level in obesity, and further studies are necessary. However, several studies have shown that DIO1 is more highly expressed in WAT and DIO_2_ in BAT and that DIO1 levels and activity in WAT are particularly higher in obese patients. Furthermore, in obese subjects, DIO1 levels are correlated with leptin expression, thus suggesting that leptin, together with DIO1 and T3, plays an important role in the local control of adipose tissue metabolism, providing at least part of an explanation for the increase in free T3/free T4 ratio in obese patients [66]. Also at the central level, leptin exerts an important role in the regulation of the hypothalamus–pituitary–thyroid axis and is also believed to be associated with the progression of some thyroid cancers [67].

The WAT is a reservoir of lipids in the form of triglycerides, and in normal human subjects it accounts for about 15% of total body weight, reaching up to 40% in obese conditions. WAT is also considered an organ with important endocrine functions, that produces adipocytokines, which are various bioactive peptides/proteins, immune and inflammatory molecules, important in energy homeostasis, appetite, glucose and lipid metabolism, insulin sensitivity, inflammation, etc. Adipocytokines influence target tissues (i.e., liver, pancreas, skeletal muscle, central nervous system, heart, vessels, etc.) through endocrine, autocrine, and paracrine effects [68]. Depending on its morphology, function, and anatomical distribution, WAT can be subdivided into subcutaneous and visceral (mesenteric, omental, retroperitoneal, epicardial, and perivascular) fat. WAT can change volume, expanding or contracting, and its growth occurs by hyperplasia, consisting in an increased number of fat cells, and hypertrophy, that is an increased adipocyte volume. The size of adipocytes depends on their triglyceride content, resulting from the balance between processes of lipogenesis, lipolysis, and fatty acid oxidation, strictly under insulin, catecholamines regulation, but also under TH control [69]. THs also regulate the adipogenesis process, which consists in the generation of new adipocytes, starting from progenitor cells, and requires the presence and activity of some transcription factors, such as peroxisome proliferator-activated receptor (PPARγ) and CCAAT/enhancer-binding proteins (C/EBPs) [70]. The BAT has been found in dispersed sites in different organs (heart, kidneys, etc.), and its cells are characterized by small droplets and a high number of mitochondria, especially at cold temperatures [70]. BAT presides over cold adaptation and thermogenesis by energy dissipation, which makes this tissue an interesting potential tool against obesity development. At the BAT level, locally produced T3 and norepinephrine (NE) released from the sympathetic nerve synergistically interact to heat production, and the process requires the involvement of DIO_2_ and uncoupling protein-1 (UCP-1), present in the inner membrane of mitochondria [71]. BAT activation starts when NE, binding to adrenergic receptors, induces lipolysis activation via adenylate cyclase and cAMP increase [72]. Lipolysis brings augmented levels of fatty acids which, in turn, stimulate UCP-1 which, by uncoupling the respiratory chain, stimulates heat production [73]. T3 further amplifies the adrenergic effects in BAT by a direct action on UCP1 transcription and mRNA stabilization. In fact, the UCP1 promoter has thyroid response elements (TREs), important for direct interaction of TR complex [74]. In BAT, adrenergic stimulation and cold exposure induce the activation of DIO_2_ and the consequent increase in T4 to T3 conversion, required for a full thermogenic response [65]. In DIO_2_-KO mice has been observed the induction of an excessive adrenergic stimulation aiming to counteract the absence of T3 normally produced by DIO_2_ activity. However, no efficient levels of fatty acids were produced to adequately respond to cold exposure, confirming the relevant role of DIO_2_ in normal lipogenesis process [75]. Stimulation of thermogenesis in BAT is characterized by a reduction in body weight, increased insulin sensitivity, and uptake of lipids. In certain conditions in which a thermogenic phenotype is required, white adipocytes may undergo the so-called process of “browning”, in which adipocytes with a beige color and containing UPC1 make their appearance and have different molecular and functional properties with respect to brown adipocytes [76]. Under basal conditions, beige adipocytes express low UCP1, but the levels augment upon induction in response to cold, promoting thermogenesis and energy expenditure [77]. In experimental animals, it was observed that the return of beige adipocytes to warmer conditions may induce the loss of UCP1 expression, suggesting that the processes involved may be reversible. The leptin acts together with insulin on proopiomelanocortin neurons in the central nervous system to activate the WAT browning process [78], whereas at the peripheral level, it was proposed that, at least in part, leptin can stimulate browning through the regulation of irisin expression, a myokine produced in skeletal muscle in response to intense exercise. However, more data are needed in this direction [79]. Interestingly, the levels of leptin secreted by adipocytes are stimulated by TSH receptor-binding, whereas, in turn, leptin stimulates the intracellular T3 synthesis, modulating deiodinase activity in the adipocyte, evidencing a close TSH/leptin positive feedback. Thus, obesity is characterized by elevated leptin levels, which correlate with raised TSH, and decreased FT4 values, enhancing the susceptibility to thyroid autoimmunity and subsequent hypothyroidism. Consequently, in a subject with obesity, TSH levels may be just a functional consequence of obesity, as well as dysthyroidism, especially in its subclinical expression [80,81]. Recent studies indicated that also human BAT might derive from conversion of beige/brite adipocytes into cells with a brown-like phenotype and that this process may have potential beneficial metabolic consequences. Even though in humans the browning events have been described also as a secondary effect of some pathophysiological conditions, some other studies indicated that, in humans, the browning process might have important physiologic relevance, in response to change of season and cold exposure. Data on browning in humans are still scarce; however, several browning agents have been identified in recent years, and some of them are currently being investigated in humans. A better understanding of the role of BAT in human metabolism and its interrelationship with body fat distribution and energy expenditure, by either increasing functional BAT or inducing white adipose browning, is very attractive in the perspective of identifying new therapeutic strategies for the treatment of obesity and associated metabolic disorders.

## 5. TH, Hypertension, and Cardiometabolic Risk

Hypertension is one of the main components of MetS and also one of the main risk factors of CV mortality and morbidity [82]. In this respect, PAMELA study indicates that over 80% of MetS patients exhibit hypertension [83]. The relationship between hypertension and MetS is due to insulin resistance, central or visceral obesity, sympathetic hyperarousal, activation of the renin–angiotensin system, oxidative stress, and raised inflammatory factors [84]. For insulin resistance, there is an increase in sympathetic nervous system activity, reduction in NO synthesis, with sodium retention as consequence of obesity, thus exacerbating hypertension. Other factors, such as leptin action, obstructive sleep apnea, and baroreflex dysfunction, are involved. Hypertension is often associated with TH diseases [85]. In hyperthyroidism, patients are characterized by an increase in heart rate, pulse amplitude, and cardiac output, simulating a growth adrenergic activity with, however, normal levels of cathecolamines. T3 excess causes dilation of resistance arterioles, reducing systemic vascular resistance. This stimulates renin release, sodium reabsorption, and blood volume expansion. The co-occurrence of ischemic and hypertensive condition in the hyperthyroid patients may affect the heart capacity to reply to the changes in metabolic needs [86]. In hyperthyroidism, T4 increases arterial stiffness due to its effects on smooth muscle and endothelial cells [87]. In elderly patients with atherosclerosis, this can lead to a significant increase in systolic blood pressure. Additionally, T3 stimulates erythropoietin synthesis, leading to an increase in red blood cell mass, blood volume, and cardiac preload [88]. Also, the condition of subclinical hyperthyroidism (SHyper), characterized by subnormal serum TSH levels and T4 and T3 within normal range, in the long term, leads to changes in cardiac function, consisting of impairment in diastolic function due to reduced myocardial relaxation, and occurrence of arrhythmias such as atrial fibrillation [89]. Moreover, endothelial dysfunctions have been found [90]. Nonetheless, no clear association with hypertension has been described [91,92,93]. In fact, whereas Kaminski and colleagues have reported a higher nocturnal mean systolic and diastolic blood pressure and higher mean blood pressure in patients with subclinical hyperthyroidism compared to control subjects, the results from Pomerania Study showed no association [94,95]. In the case of hypothyroidism, the clinical condition characterized by an elevated TSH, combined with reduced circulating free-T4 and free-T3 levels, has been reported an association with hypertension in about 3% of patients, although this relation is often overlooked. According to the World Health Organization, there is a 3-fold higher prevalence of hypertension in hypothyroidism, probably caused by an increase in systemic vascular resistance [96]. Several studies have indicated that elevated levels of TSH predict moderately higher blood pressure, with a 1 mU/l increase in TSH associated with an approximately 2 mmHg increase in systolic pressure and a 1–2 mmHg increase in diastolic pressure. Furthermore, many lines of evidence have also documented endothelial dysfunction and dyslipidemia [97,98], as indicated by an impaired artery flow-mediated dilation and increased carotid intima-media thickness. One of the possible mechanisms involved is the reduction in NO. In fact, in physiological condition, THs stimulate endothelial NO production via non-genomic actions by activating the PI3K and the serine/threonine-protein kinase signaling pathways [99]. In subjects with hypothyroidism, low levels of NO may cause endothelial dysfunction with consequences on vascular tone modulation and contributing to atherosclerosis formation [100]. In SCH patients, there is an increase in systemic vascular resistance, diastolic dysfunction, an impairment in ventricular filling, and relaxation [101]. Also, a decreased endothelium-dependent vasodilatation in the aorta and renal blood vessels, with consequent reduction in the activity of endothelial NO synthase (eNOS, the constitutive enzymes for NO production), was established [102].

## 6. Inflammation and Oxidative Stress and Endothelial Dysfunction: A Common Underlying Mechanism of Dysthyroidism, Cardiovascular Risk, and Metabolic Syndrome

Oxidative stress is a status characterized by an imbalance between the production of free radicals and the antioxidant system’s capacity to oppose their actions. Reactive oxygen species (ROS) and nitrogen species contribute to cellular dysfunction during oxidative stress, damaging proteins, lipid membranes, and nucleic acids, leading to cell death. ROSs also act as second messengers, influencing the modulation of vascular redox status and platelet thrombus formation [103]. Various mechanisms contribute to the generation of oxygen-free radicals, with glucose oxidation considered one of the primary sources [104]. If not neutralized, superoxide anionic radicals can form highly reactive hydroxyl radicals or can react with NO to form peroxynitrite, promoting the lipid peroxidation of LDL. In hyperglycemic conditions, the interaction of glucose with proteins leads to the formation of advanced glycation end products (AGEs), contributing to intracellular oxidative stress. AGEs activate nuclear factor-kappa-light-chain-enhancer of activated B cells (NF-κB), supporting the TNF-α-induced inflammatory response [105]. Modulating the NF-κB pathway or specific downstream genes may therefore represent an approach to control or prevent chronic inflammatory diseases such as atherosclerosis. Endothelial dysfunction, the initial stage of atherosclerotic plaque formation, develops in response to the inflammatory/oxidative stress-related stimulus [106]. This condition is also seen as a potential mechanism in the negative consequences of MetS. In this context, the level of inflammation may indicate an exceptionally high risk of adverse outcomes, with inflammation resulting from increased oxidative stress through LDL oxidative modification [107]. Modified LDL, caused by oxidation, can lead to the pathogenic generation of atherosclerotic plaque, releasing pro-inflammatory mediators and triggering a chronic inflammatory reaction, causing leukocyte recruitment, foam cell formation, and inflammation [108]. During chronic vascular inflammatory conditions, endothelial cells express various inflammatory proteins and cytokines, with NF-κB activation leading to the production of cytokines such as IL-6 and IL-8. The balance between pro- and anti-inflammatory cytokines regulates the inflammatory response and thrombus formation; additionally, during chronic vascular inflammatory processes, various inflammatory markers are considered potential indicators for predicting coronary events [109]. In summary, inflammation plays a central role in the development and progression of atherosclerosis, and understanding it is crucial for identifying at-risk patients and developing targeted therapeutic strategies. Evidence has demonstrated the effects of thyroid diseases on oxidative stress and inflammation. In fact, T3 modulates mitochondrial function at different tissue sites, (e.g., skeletal muscle, heart, kidney, and liver) [110]. Accordingly, hyperthyroidism has been associated with increased oxidative stress and reduction in the antioxidants’ defenses [111]. In hypothyroidism, the metabolic rate slows so that the oxidative stress is expected to decrease [112]. Nonetheless, different data showed an increase in oxidative stress, which may be improved by levothyroxine replacement [111,113,114,115]. Moreover, elevated levels of cytokines (e.g., IL6, IL10, IL17, and TNF-α) and changes in NO availability were found in patients with overt or subclinical hypothyroidism [116,117,118]. THs modulate expression of uncoupling proteins in the mitochondria of fat and skeletal muscle, adrenergic receptor numbers by enhancing responsiveness of catecholamines, and thus regulate metabolic rate and body weight [119]. Moreover, since obesity, glucose, lipids, and blood pressure are all MetS components, THs represent a useful predictive biomarker for risk to develop MetS [29]. A bidirectional association between free-T3 and MetS and its components likely exists as MetS (or its components) leads to thyroid dysfunction, whereas thyroid alterations may cause a higher incidence of MetS in adults [13]. As an example, THs affect metabolism, in terms of weight gain and adiposity, also influencing core body temperature, appetite, and sympathetic activity (underlying mechanisms involved include adenosine triphosphate-ATP utilization, uncoupling synthesis of ATP, mitochondrial biogenesis, and the inotropic and chronotropic events). Conversely, thyroid function is influenced by adiposity, for example, by leptin, but also pro-inflammatory cytokines related to obesity and insulin resistance [95]. In particular, leptin stimulates the hypothalamus–pituitary–thyroid axis, regulates the thyroid receptor gene expression, and is involved in the activation of T4 to T3 by deiodinase enzymes, causing mild elevation in TSH, and increasing TH levels [120]. T3, in turn, affects lipids by modulating their synthesis, mobilization, and degradation, as well as mediating the expression of genes for key enzymes involved in lipid metabolism (e.g., hydroxymethyl glutaryl coenzyme); T3 can also stimulate hepatic glucose production via direct actions (liver) or indirectly via a sympathetic pathway (hypothalamus), whereas thyroid dysfunction (hyperthyroidism and hypothyroidism) may affect glucose metabolism through actions on different organs (e.g., gastrointestinal tract, pancreas, adipose tissue, skeletal muscles, and the central nervous system) [121,122]. TH dysfunction may induce hypertrophy, increased heart rate, and altered contractility. Moreover, overexpression of cardiac DIO_2_, the main DIO isoform in the heart, has been found to improve contractile function, restore heart function, and normalize the expression of several genes involved in pathological remodeling in mouse hearts after pressure overload [123].

Selenium (Se), a component of selenoproteins, is essential as an antioxidant and regulator of TH metabolism [124]. Increasing data show the relationship between Se and selenoproteins and glucose metabolism, insulin resistance, and diabetes and its complications [125,126]. Se supplementation may improve lipid profile (e.g., decreasing total cholesterol levels) and modulate oxidative damage, inflammation, energy balance, vascular cell apoptosis, calcification, endothelial dysfunction, and heart function, supporting the hypothesis that an adequate Se intake can exert athero-protective actions mediated by the effects of selenoproteins, likely including selenocysteine-containing DIO isoforms, regulating TH metabolism [127,128]. However, there is still no shared consensus on the role of Se blood levels as an effective biomarker in the pathophysiology and monitoring nor the beneficial effect of Se supplementation of cardiometabolic disease [129,130]. In any case, targeting TH-induced oxidative stress could represent an attractive additive interventional tool in the future, and the importance of this aspect is underlined by the fact that restoration of normal thyroid function may improve cardiometabolic risk [131,132]. An interesting aspect is the emergence of several interrelated biomarkers, recently discovered to be involved in the relationship between TH, MetS, and CV disease, which can better elucidate the pathophysiology of this triad and serve as tools for a better risk stratification and monitoring of disease development.

It is the case of irisin, vitamin, and ceramides; irisin is an adipo-myokine also expressed in thyroid tissue, with effects on metabolism and thermogenesis (thyroid can be directly/indirectly modulated by irisin, as well as irisin can affect thyroid) and found correlated to CV disease, MetS, and obesity, reflecting a protective compensatory mechanism against oxidative muscle and thyroid cell stress; vitamin D, mainly produced by skin exposure to ultraviolet irradiation, and the main regulator of calcium homeostasis and bone health, shows antioxidant properties and is involved in cardiometabolic conditions; ceramides are lipids and structural building blocks for the genesis of other more complex sphingolipid molecules, mediators of inflammation and oxidative stress and related to cardiometabolic diseases [133,134,135,136]. Vitamin D supplementation has been found to modulate irisin concentration in subjects with vitamin D deficiency and in T2DM patients [137,138]. Moreover, vitamin D supplementation (50,000 IU/week for 12 weeks) improves serum irisin together with TSH, total cholesterol levels, and body composition in women with SCH [139]. Previous in vitro studies on human endothelial cells suggested one mechanism by which vitamin D can affect irisin levels: the vitamin D-related activation of the silent information regulator sirtuin 1 (SIRT1), AMP-activated protein kinase (AMPK), and peroxisome proliferator-activated receptor gamma coactivator 1-alpha (Pgc1α), enhancing expression of irisin precursor FNDC5 and increase in irisin levels [140,141]. Vitamin D can also regulate sphingolipid metabolism (e.g., by modulating the hydrolysis of sphingomyelin, S1P receptors, and sphingosine kinase 1 and 2 expression, downregulating ceramide kinase, and upregulating ceramide [142]. Accordingly, data from experimental and clinical studies indicated that vitamin D supplementation modulates sphingolipid levels in type 2 diabetes, dyslipidemia, and overweight/obese subjects [142]. The relationship is mutual, as some results have also evidenced how sphingolipid localization and expression in lipid rafts (membrane lipid-rich microdomains) modify vitamin D receptor localization and functioning [142]. Interestingly, recent data evidenced that irisin is a target of the Sphingosine-1-Phosphate/Sphingosine-1-Phosphate receptor-mediated (S1P/S1PR) axis in murine skeletal muscle cells; conversely, irisin modulates S1PR expression (reduction) the expression of Sphingosine-1-phosphate receptor 2, thus suggesting the existence of a functional link between the S1P/S1PR and irisin signaling pathways [143,144].

## 7. Conclusions

THs interfere with each component of MetS, from hypertension to insulin resistance, lipid regulation, and adipose tissue. This role takes place at the molecular level and is reflected at the clinical level with an increased risk of CV events (see Graphical Abstract).

Thus, MetS is frequently associated with thyroid dysfunction, which supports the need to assess thyroid function in this group, and when clinically indicated, to correct it to maintain euthyroidism [62]. However, there are still several critical points to be further investigated both at the molecular and clinical level, in particular considering the need to treat subclinical dysthyroidism in MetS patients.

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
