# Peer review of "Metabolic Syndrome, Thyroid Dysfunction, and Cardiovascular Risk: The Triptych of Evil"

_ijms, 2024, doi:10.3390/ijms251910628_

Round 1

Reviewer 1 Report

Comments and Suggestions for Authors

In the introduction the authors should also mention the negative impact  of metabolic syndrome on patients with covid 19, they may refer to 10.3390/biomedicines10071519 and 10.7759/cureus.27438

In the part of the introduction where the authors are mentioning TH, they should expand on these and clearly mention the types of TH and their effects specifically, also they should mention the current guidelines when it comes to commencing treatment with TH agonists and/or antagonists, depending on the situation.

Moreover, the authors should also mention potential causes of thyroid dysfunction, including iatrogenic ones such as the use of amiodarone, since the manuscript also focuses on CV disease.

In my opinion the authors should also focus on the cardiac an vascular receptors of TH, highlighting their importance in this space.

The authors mention in lines 138 to 139 the role of modulatory biomarkers but again do not accentuate the role of different TH receptors in lipid metabolism.

The authors use the term browning when probably referring to some sort metaplasia of adipose tissue, however is this truly a scientific term ? because it is quite scarce in literature.

Authors have cited several references over 25 years old, with the current knowledge available especially on such a broad subject I consider that the authors should find more current articles.

Another aspect is that the authors seem to have relied heavily on previous work when constructing this manuscript. 

Comments on the Quality of English Language

English is fine overall. 

Author Response

In the introduction the authors should also mention the negative impact  of metabolic syndrome on patients with covid 19, they may refer to 10.3390/biomedicines10071519 and 10.7759/cureus.27438

We added the following sentences and references: “In this context, the intermingled relationship between TH, cardiometabolic risk is highlighted also in the recent SARS-CoV-2 pandemic; in fact, evidence emerged that preexisting traditional cardiovascular risk factors may negatively impact the risk and severity of COVID-19

Makhoul E, Aklinski JL, Miller J, Leonard C, Backer S, Kahar P, Parmar MS, Khanna D. A Review of COVID-19 in Relation to Metabolic Syndrome: Obesity, Hypertension, Diabetes, and Dyslipidemia. Cureus. 2022 Jul 29;14(7):e27438.

Guzik TJ, Mohiddin SA, Dimarco A, Patel V, Savvatis K, Marelli-Berg FM, Madhur MS, Tomaszewski M, Maffia P, D'Acquisto F, Nicklin SA, Marian AJ, Nosalski R, Murray EC, Guzik B, Berry C, Touyz RM, Kreutz R, Wang DW, Bhella D, Sagliocco O, Crea F, Thomson EC, McInnes IB. COVID-19 and the cardiovascular system: implications for risk assessment, diagnosis, and treatment options. Cardiovasc Res. 2020 Aug 1;116(10):1666-1687.

Tudoran C, Tudoran M, Cut TG, Lazureanu VE, Bende F, Fofiu R, Enache A, Pescariu SA, Novacescu D. The Impact of Metabolic Syndrome and Obesity on the Evolution of Diastolic Dysfunction in Apparently Healthy Patients Suffering from Post-COVID-19 Syndrome. Biomedicines. 2022 Jun 27;10(7):1519.

Vassalle C. Viral infections in cardiometabolic risk and disease between old acquaintances and new enemies. Explor Cardiol. 2023;1:148–79.

In the part of the introduction where the authors are mentioning TH, they should expand on these and clearly mention the types of TH and their effects specifically, also they should mention the current guidelines when it comes to commencing treatment with TH agonists and/or antagonists, depending on the situation.

We added the following sentences: “Triiodothyronine (T3) is the active hormone of thyroid and derives from  5’-monodeiodination of thyroxine (T4), produced directly by the thyroid gland. This process occurring in the peripheral tissues is mediated by type I and II deiodinases. T3 acts on the heart through genomic and non-genomic mechanisms. The genomic ones are mediated by thyroid receptors (TR) α1 and α2, and β1 and β2. They bind to TH response elements (TREs) in the promoter region of genes. In the heart, TRα1 is the more diffuse TR isoform and has a functional regulatory role, whereas TRα2 acts as a counter-regulator of TRα1suppressing its transcriptional effects. The non-genomic actions are mediated through cytoplasmic and membrane-associated TRs, such as integrin αVβ3 or extranuclear TRα and β, and involve ions, glucose and amino acid transport across the plasma membrane. Thus, through these mechanisms, THs exert effects on cardiac morphology and structure, coronary vasculature, cell metabolism, cell protection, growth, and differentiation”.

“With regard to the dysthyroid treatment, in the case of hypothyroidism, L-T4 substitutive treatment, that is the main one, is recommended when TSH levels are higher than 10 mIU/L, starting with a dose of 20-25 microgr/day. In the case of hyperthyroidism, it can be treated with antithyroid drugs, radioactive iodine ablation, or surgery. In a recent study, surgery was associated with lower long-term risks of cardiac events and all-cause mortality, while radioactive iodine ablation was associated with a lower incidence of cardiac events in comparison to antithyroid drugs”.

Garber JR, Cobin RH, Gharib H, Hennessey JV, Klein I, Mechanick JI, et al. Clinical practice guidelines for hypothyroidism in adults: cosponsored by the American Association of Clinical Endocrinologists and the American

Peng CC, Lin YJ, Lee SY, Lin SM, Han C, Loh CH, Huang HK, Pearce EN. JAMA Netw Open. 2024 Mar 4;7(3):e240904. doi: 10.1001/jamanetworkopen.2024.0904. MACE and Hyperthyroidism Treated With Medication, Radioactive Iodine, or Thyroidectomy.

Moreover, the authors should also mention potential causes of thyroid dysfunction, including iatrogenic ones such as the use of amiodarone, since the manuscript also focuses on CV disease.

We added the following sentences and references: “The causes of hypothyroidism can be due to autoimmune diseases, mainly Hashimoto’s thyroiditis, or infectious, or thyroid infiltration diseases, for example amyloidosis, or endemic goiter, and iatrogenic. With regard to hyperthyroidism, the main cause is Grave’s disease, whereas the other causes are toxic nodules and thyrotoxic phase of thyroiditis, and iatrogenic. Among the iatrogenic causes, amiodarone induces dysthyroidism, both hyper- or hypo-thyroidism, in 15-20% of patients with cardiac diseases. There are two forms of amiodarone-induced thyrotoxicosis; the first one characterized by a iodine-induced hyperthyroidism, the second one due to a destructive thyroiditis”. 

Lee, S.Y., Pearce., E.N. (2023). Hyperthyroidism: a review. JAMA 2023;330:1472-1483. doi:10.1001/jama.2023.19052. Giorda, C. B., Carnà, P., Romeo, F., Costa, G., Tartaglino, B., & Gnavi, R. (2017). Prevalence, incidence and associated comorbidities of treated hypothyroidism: an update from a European population. European Journal of Endocrinology, 176(5), 533–542. doi:10.1530/eje-16-0559 

Cappellani, D., Papini, P., Pingitore, A., Tomisti, L., Mantuano, M., Di Certo, A. M., … Bogazzi, F. (2019). Comparison between total thyroidectomy and medical therapy for amiodarone-induced thyrotoxicosis. The Journal of Clinical Endocrinology & Metabolism. doi:10.1210/clinem/dgz041

In my opinion the authors should also focus on the cardiac an vascular receptors of TH, highlighting their importance in this space.

With regard to the TH receptors, please, look at the previous question

The authors mention in lines 138 to 139 the role of modulatory biomarkers but again do not accentuate the role of different TH receptors in lipid metabolism.

We added a paragraph explaining the TH receptors and the  link to the lipid metabolism.

 “THs exert their actions by binding to thyroid hormone receptors (TRs) influencing all major metabolic pathways with specific regard to lipid metabolism, affect synthesis, mobilization and, even more, degradation of lipids, although degradation is influenced more than synthesis. In particular, TRs are encoded by two genes, THRA and THRB, which generate  subtypes α and β (e.g. TRα1, TRα2, TRβ1, and TRβ2). TRα is crucial for heart rate and for cardiac contractility and relaxation, whereas TRβ1, the major TR in the liver, is interested in lipid metabolism. TH derivatives that bind TRβ selectively or exhibit liver-selective uptake improve plasma lipids without affecting heart rate.  TRβ1 resulted the most mediator of the effects of T3 on cholesterol and lipoprotein metabolism in mice.  In several animal studies, treatment with TRβ1 selective TH analogs, resulted in decreased plasma TG-levels  both in hypothyroid and euthyroid mice.

Johansson L, Rudling M, Scanlan TS, Lundasen T, Webb P, Baxter J, et al. Selective thyroid receptor modulation by GC-1 reduces serum lipids and stimulates steps of reverse cholesterol transport in euthyroid mice. Proc Natl Acad Sci U S A. 2005;102(29):10297–302

  1. Gullberg, M. Rudling, C. Salto, D. Forrest, B. Angelin, B. Vennstrom Requirement for thyroid hormone receptor beta in T3 regulation of cholesterol metabolism in mice Mol. Endocrinol., 16 (2002), pp. 1767-1777

S.U. Trost, E. Swanson, B. Gloss, D.B. Wang-Iverson, H. Zhang, T. Volodarsky, G.J. Grover, J.D. Baxter, G. Chiellini, T.S. Scanlan, W.H. DillmannThe thyroid hormone receptor-beta-selective agonist GC-1 differentially affects plasma lipids and cardiac activity Endocrinology, 141 (2000), pp. 3057-3064)

  1. Johansson, M. Rudling, T.S. Scanlan, T. Lundasen, P. Webb, J. Baxter, B. Angelin, P. Parini Selective thyroid receptor modulation by GC-1 reduces serum lipids and stimulates steps of reverse cholesterol transport in euthyroid mice Proc. Natl Acad. Sci. USA, 102 (2005), pp. 10297

The authors use the term browning when probably referring to some sort metaplasia of adipose tissue, however is this truly a scientific term ? because it is quite scarce in literature.

Browning” is a scientific term used to describe a transformation of beige (or brite) adipocytes, present in white adipose tissue. In the basal state, beige/brite adipocytes are characterized by a white fat-like phenotype, however, under certain conditions, they acquire a brown fat-like phenotype, leading to increased thermogenesis. Recent studies indicate that also human brown adipose tissue, present in some specific locations in the human body, might derive from conversion of beige/brite adipocytes into cells with a brown-like phenotype, with potential beneficial metabolic consequences. It is true that the browning in humans has been observed also as secondary effect of some pathophysiological condition, but several studies indicate that the browning process might have a physiologic relevance in humans, in response to change of season and cold exposure. We agree with the reviewer about the scarcity of data on browning on humans, however this process is very attractive in the treatment of some disorders, especially the ones associated to obesity, and we believe that is worth it to be mentioned in this review, given the regulatory role of THs on adipose tissue.

We added the following sentences: “Recent studies indicated that also human BAT might derive from conversion of beige/brite adipocytes into cells with a brown-like phenotype, and that this process may have potential beneficial metabolic consequences. Even though in humans the browning events have been described also as secondary effect of some pathophysiological conditions, some other studies indicated that, in humans, the browning process might have important physiologic relevance, in response to change of season and cold exposure. Data on browning in humans are still scarce, however, several browning agents have been identified in recent years and some of them are currently being investigated in humans. A better understanding of the role of BAT in human metabolism and its interrelationship with body fat distribution and energy expenditure, by either increasing functional BAT or inducing white adipose browning, is very attractive in the perspective of identifying new therapeutic strategies for treatment of obesity and associated metabolic disorders”.

Authors have cited several references over 25 years old, with the current knowledge available especially on such a broad subject I consider that the authors should find more current article.

In the old version of the manuscript, there were 6 references over 25 years (2,3,47,49,51 and 64). We deleted reference 2 and 3, changed reference 64; we left the other 3 references because we specifically referred to the results of the mentioned studies.

Another aspect is that the authors seem to have relied heavily on previous work when constructing this manuscript. 

Accordingly, we modified some sentences in the revised manuscript.

Reviewer 2 Report

Comments and Suggestions for Authors

Difficult paper without clear massage for the reader, and without clear conclusion. As thyroid dysfunction corectable disfunction it would be nice to show when therapy is needed. What we can obtain by treating thyroid dysfunction in MET.

Thyroid dysfunction cannot be considered as a predictive biomarker of MET! The CV risk related to thyroid dysfunction increase the risk outside MET!

Thyroid hormones regulate the rate of carbohydrate and lipid metabolism, but not their metabolism! The role of thyroid hormone in the regulation of BP is marginal outside of severe disturbances.

The paper did not review the epidemiological data concerning thyroid dysfunction in MS. It should be addedd. In addition please note that high leptin levels by stimulation of TSH secretion may make the diagnosis of subclinical hypothyroidism chalanging.

Furthermore hypothyroidism is easily correctable, much more than obesity. The assosiation between subclinical hypothyroidism and CV concerning the mechanisms is debatable, and listed mechanisms (line 76-78) are only proven for severe disturbances.

Are there any studies analysing insulin resistance with clamp technique in thyroid disturbances? The HOMA-IR model has some limitations.

BAT in adult human has vary limited role. Was 'browning' documented in humans?

Adipipose tissue is not an endocrine organ, but organ with endocrine function!!!

Please note that hyperthyroidism is curable transient clinical condition. Thjerefore long-lasting conseqences are overestimated.

I have dysagree that 'TH has a key role to maintain CV homeostasis.' 

'homeostatic cardiovascular role" of thyroid hormones - not acceptable.

'It is noteworthy that in the experimental setting of acute myocardial infarction, TH administration exerts a cardioprotective effect on the myocardium, including antiapoptotic and antifibrotic effect, mitochondrial protection, cell growth and differentiation, and neoangiogenesis [16,32].' very controversial in summary.

There is no clear massage comming from conclusions.

Table 1 and Figure 1 are missing

Minore:

line 35 'leukocytes'???

Author Response

Reviewer 2

Difficult paper without clear massage for the reader, and without clear conclusion. As thyroid dysfunction corectable disfunction it would be nice to show when therapy is needed. What we can obtain by treating thyroid dysfunction in MET.

We changed the conclusion. First, we have moved the following sentences from the conclusion to the introduction: “Triiodothyronine (T3) is the active hormone of thyroid and derives from  5’-monodeiodination of thyroxine (T4), produced directly by the thyroid gland. This process occurring in the peripheral tissues is mediated by type I and II deiodinases. T3 acts on the heart through genomic and non-genomic mechanisms. The genomic ones are mediated by thyroid receptors (TR) α1 and α2, and β1 and β2. They bind to TH response elements (TREs) in the promoter region of genes. In the heart, TRα1 is the more diffuse TR isoform and has a functional regulatory role, whereas TRα2 acts as a counter-regulator of TRα1suppressing its transcriptional effects. The non-genomic actions are mediated through cytoplasmic and membrane-associated TRs, such as integrin αVβ3 or extranuclear TRα and β, and involve ions, glucose and amino acid transport across the plasma membrane. The TH effects on CV system are also indirect through the regulation of hormonal or neuroendocrine pathways [16]. Thus, through these mechanisms, THs exert effects on cardiac morphology and structure, coronary vasculature, cell metabolism, cell protection, growth, and differentiation. The important CV role of TH is evident since mild thyroid dysfunctions (subclinical hypothyroidism or hyperthyroidism) induces changes in cardiac function and morphology [120] and are associated with increased CV morbidity and mortality [121-127]. In a recent study, TH metabolic abnormalities increase CV risk in the general population [128]. Accordingly, the 10-year absolute CV risk increased over 5% for women with free-T4 greater than the 85th percentile (median 17.6 pmol/L) and men with free-T4 greater than the 75th percentile (median 16.7 pmol/L). Similarly, low TSH circulating levels were associated with a higher risk of all-cause and CV mortality. In the context of heart failure and acute myocardial infarction, abnormal TH profile has been associated with a high prevalence of major cardiac event, representing an independent predictor of cardiac death in addition to the common clinical and cardiac functional parameters [129,130]”.

The conclusion is the following: “TH interferes on each component of METs, from hypertension to insulin resistance, lipid regulation and adipose tissue. This role takes place at the molecular level and is reflected at the clinical level with an increased risk of CV events.  This  highlights the usefulness to assess thyroid metabolic function in subjects with METs in order to provide therapeutic interventions to maintain euthyroidism that is easily correctable. [131]”.

Thyroid dysfunction cannot be considered as a predictive biomarker of MET! The CV risk related to thyroid dysfunction increase the risk outside MET!

Accordingly, we deleted the sentence

Thyroid hormones regulate the rate of carbohydrate and lipid metabolism, but not their metabolism! The role of thyroid hormone in the regulation of BP is marginal outside of severe disturbances.

Accordingly, we changed the sentence as follows: “Thyroid hormones (TH) play a key role to regulate the rate of carbohydrate and lipid metabolism, inducing also abnormalities of blood pressure values in the presence of thyroid metabolism disturbances”.

The paper did not review the epidemiological data concerning thyroid dysfunction in MS. It should be addedd. In addition please note that high leptin levels by stimulation of TSH secretion may make the diagnosis of subclinical hypothyroidism chalanging.

Accordingly, we added the following sentences and references: “In the clinical setting, thyroid function within normal ranges was associated to METs, in which higher free-thyroxine (T4) levels were associated to a low risk of METs [14]. However the relationship between METs and TH disorders is still not clearly defined and the current results are contrasting. Accordingly, a recent meta-analysis did not rise any conclusion to high heterogeneity in reporting results Another study showed that each unit increase in TSH was associated with a 3% increase in the odds of prevalent METs. When considering subclinical hypothyroidism (SCH) with a TSH > 10 mIU/l, an increased odds of prevalent but not incident METs has been shown [15]. In addition, an age factor has been shown by Wu et al with the evidence of a relationship of SHYPO and METs in young men”.

Alwan H, Aponte Ribero V, Efthimiou O, Del Giovane C, Rodondi N, Duntas L. A systematic review and meta-analysis investigating the relationship between metabolic syndrome and the incidence of thyroid diseases. Endocrine. 2024;84:320-327. doi: 10.1007/s12020-023-03503-7.

Wu Z, Jiang Y, Zhou D, Chen S, Zhao Y, Zhang H, Liu Y, Li X, Wang W, Zhang J, Kang X, Tao L, Gao B, Guo X. Sex-specific Association of Subclinical Hypothyroidism With Incident Metabolic Syndrome: A Population-based Cohort Study. J Clin Endocrinol Metab. 2022;107:e2365-e2372. doi: 10.1210/clinem/dgac110.

With regard to leptin and TSH, we wrote: “Interestingly, the levels of leptin secreted by adipocytes are stimulated by TSH receptor-binding, whereas in turns, leptin stimulates the intracellular T3 synthesis, modulating deiodinase activity in the adipocyte, evidencing a close TSH/leptin positive feedback. Thus, obesity is characterised by elevated leptin levels, which correlated with raised TSH, and decreased FT4 values, enhancing the susceptibility to thyroid autoimmunity and subsequent hypothyroidism. Consequently, in a subject with obesity, TSH may be just a functional consequence of obesity, as well as dysthyroidism, especially in its subclinical expression”.

Biondi B. Subclinical Hypothyroidism in Patients with Obesity and Metabolic Syndrome: A Narrative Review. Nutrients. 2023 Dec 27;16(1):87.

Sanyal D, Raychaudhuri M. Hypothyroidism and obesity: An intriguing link. Indian J Endocrinol Metab. 2016 Jul-Aug;20(4):554-7.

Furthermore hypothyroidism is easily correctable, much more than obesity. The assosiation between subclinical hypothyroidism and CV concerning the mechanisms is debatable, and listed mechanisms (line 76-78) are only proven for severe disturbances.

Accordingly, we introduced the following sentence: “These effects are much more evident in the presence of severe thyroid disturbances and are reversible when euthyroidism is restored”.

Are there any studies analysing insulin resistance with clamp technique in thyroid disturbances? The HOMA-IR model has some limitations.

In the review, we already included the current reference 21 in which HOMA-IR has been applied in SCH patients. In addition we added the reference Chubb SAP, Davis WA, Davis TME. Interactions among thyroid function, insulin sensitivity, and serum lipid concentrations: the Fremantle diabetes study.  J Clin Endocrinol Metab. 2005 Sep;90(9):5317-20. doi: 10.1210/jc.2005-0298, and added the following sentence: “Moreover, Chubb et al showed that the interaction between thyroid function and insulin sensitivity is an important contributor to diabetic dyslipidemia”.

 We agree that HOMA-IR has some limitations, but we retain that this issue is out of the aims of this review.

BAT in adult human has vary limited role. Was 'browning' documented in humans?

We added the following sentences: “Recent studies indicated that also human BAT might derive from conversion of beige/brite adipocytes into cells with a brown-like phenotype, and that this process may have potential beneficial metabolic consequences. Even though in humans the browning events have been described also as secondary effect of some pathophysiological conditions, some other studies indicated that, in humans, the browning process might have important physiologic relevance, in response to change of season and cold exposure. Data on browning in humans are still scarce, however, several browning agents have been identified in recent years and some of them are currently being investigated in humans. A better understanding of the role of BAT in human metabolism and its interrelationship with body fat distribution and energy expenditure, by either increasing functional BAT or inducing white adipose browning, is very attractive in the perspective of identifying new therapeutic strategies for treatment of obesity and associated metabolic disorders”.

Adipipose tissue is not an endocrine organ, but organ with endocrine function!!!

We corrected specifying that “WAT is also considered an organ with important endocrine functions”

Please note that hyperthyroidism is curable transient clinical condition. Thjerefore long-lasting conseqences are overestimated.

We included According to the above question, we wrote that: “These effects are much more evident in the presence of severe thyroid disturbances and are reversible when euthyroidism is restored”.

I have dysagree that 'TH has a key role to maintain CV homeostasis.' 

'homeostatic cardiovascular role" of thyroid hormones - not acceptable.

We deleted the sentence.

We also added the following sentences, as suggested by the reviewer 1: “Triiodothyronine (T3) is the active hormone of thyroid and derives from  5’-monodeiodination of thyroxine (T4), produced directly by the thyroid gland. This process occurring in the peripheral tissues is mediated by type I and II deiodinases. T3 acts on the heart through genomic and non-genomic mechanisms. The genomic ones are mediated by thyroid receptors (TR) α1 and α2, and β1 and β2. They bind to TH response elements (TREs) in the promoter region of genes. In the heart, TRα1 is the more diffuse TR isoform and has a functional regulatory role, whereas TRα2 acts as a counter-regulator of TRα1suppressing its transcriptional effects. The non-genomic actions are mediated through cytoplasmic and membrane-associated TRs, such as integrin αVβ3 or extranuclear TRα and β, and involve ions, glucose and amino acid transport across the plasma membrane. Thus, through these mechanisms, THs exert effects on cardiac morphology and structure, coronary vasculature, cell metabolism, cell protection, growth, and differentiation”.

'It is noteworthy that in the experimental setting of acute myocardial infarction, TH administration exerts a cardioprotective effect on the myocardium, including antiapoptotic and antifibrotic effect, mitochondrial protection, cell growth and differentiation, and neoangiogenesis [16,32].' very controversial in summary.

Accordingly, we deleted the sentence.

We also deleted the sentence: It is noteworthy that in the experimental setting of acute myocardial infarction, TH administration exerts a cardioprotective effect on the myocardium, including antiapoptotic and antifibrotic effect, mitochondrial protection, cell growth and differentiation, and neoangiogenesis [16,32].

There is no clear massage comming from conclusions.

As we have written above, we concluded as follows:

TH interferes on each component of METs, from hypertension to insulin resistance, lipid regulation and adipose tissue. This role takes place at the molecular level and is reflected at the clinical level with an increased risk of CV events.  This highlights the usefulness to assess thyroid metabolic function in subjects with METs in order to provide therapeutic interventions to maintain euthyroidism that is easily correctable. [131]”.

Table 1 and Figure 1 are missing

We made a mistake, and deleted table 1, and added figure 1.

Minore:

line 35 'leukocytes'???

Virtually every leukocyte class – white blood cells –  (e.g. neutrophils, lymphocytes, monocytes) is implicated in atherosclerosis plaque formation, development  and its complications, and their action is neither uniform nor hierarchical (Swirski FK, Nahrendorf M. Leukocyte behaviour in atherosclerosis, myocardial infarction, and heart failure. Science. 2013 Jan 11;339(6116):161-6. )

Round 2

Reviewer 1 Report

Comments and Suggestions for Authors

The authors have responded to all of my comments accordingly. 

Comments on the Quality of English Language

English is fine overall with minor corrections needed. 

Author Response

The authors have responded to all of my comments accordingly. 

Comments on the Quality of English Language

English is fine overall with minor corrections needed. 

Accordingly we reviewed English language

Reviewer 2 Report

Comments and Suggestions for Authors

The authors have improved the paper.

I still did not see the figure and table in the manuscript.

The abstract should be corrected and include the conclusion.

Before sending the new version please accept all previous corrections. 

Author Response

The authors have improved the paper.

I still did not see the figure and table in the manuscript.

 The figure is embedded in the text and the graphical abstract uploaded as adjunctive file.

The abstract should be corrected and include the conclusion.

Accordingly, we included in the abstract the following sentence:

“This topic is critical as underlines the usefulness to assess thyroid metabolic function, actually easily correctable, in subjects with MetS in order to provide therapeutic interventions to maintain euthyroidism .”.

Before sending the new version please accept all previous corrections. 

In the new version of the manuscript, only the corrections of the second revision are shown.

Round 3

Reviewer 2 Report

Comments and Suggestions for Authors

Some additional corrections are still necessary.

1. Fig 1. 'fat tissue'

2. Conclusions: The clear clinical benefits related to obtaining euthyroidism in subjects with subclinical hypothyroidism were not proven. Please modify the conclusion. (Both in the abstract and the paper). There are even papers showing that patients with subclinical hypothyroidism are living longer.

In my opinion, the conclusion should state that metS is frequently associated with thyroid dysfunction, which supports the need to assess thyroid function in this group. Still, clear evidence supporting the treatment of subclinical hypothyroidism, regardless of pathophysiological grounds linking thyroid function with components of MetS, is uncertain.

Author Response

Some additional corrections are still necessary.

  1. Fig 1. 'fat tissue'

Checked and corrected

  1. Conclusions: The clear clinical benefits related to obtaining euthyroidism in subjects with subclinical hypothyroidism were not proven. Please modify the conclusion. (Both in the abstract and the paper). There are even papers showing that patients with subclinical hypothyroidism are living longer.

In my opinion, the conclusion should state that metS is frequently associated with thyroid dysfunction, which supports the need to assess thyroid function in this group. Still, clear evidence supporting the treatment of subclinical hypothyroidism, regardless of pathophysiological grounds linking thyroid function with components of MetS, is uncertain.

Accordingly, we inserted the following sentence in the conclusion of the abstract and of the manuscript.

Thus, MetS is frequently associated with thyroid dysfunction, which supports the need to assess thyroid function in this group, and when clinically indicated, to correct it to maintain euthyroidism. However, there are still several critical points to be further investigated both at the molecular and clinical level, in particular considering the need to treat subclinical dysthyroidism in MetS patients.”.